# DNA copy number motifs are strong and independent predictors of survival in breast cancer

Arne V. Pladsen [1,21], Gro Nilsen[2,21], Oscar M. Rueda [3], Miriam R. Aure[1], Ørnulf Borgan[4], Knut Liestøl[2], Valeria Vitelli[5], Arnoldo Frigessi[5], Anita Langerød[1], Anthony Mathelier [1,6], OSBREAC*, Olav Engebråten[9,10], Vessela Kristensen[1], David C. Wedge [18,19], Peter Van Loo [20], Carlos Caldas [3], Anne-Lise Børresen-Dale[1,9], Hege G. Russnes [1,8,22] & Ole Christian Lingjærde [1,2,14,22✉]

Somatic copy number alterations are a frequent sign of genome instability in cancer. A precise characterization of the genome architecture would reveal underlying instability mechanisms and provide an instrument for outcome prediction and treatment guidance. Here we show that the local spatial behavior of copy number profiles conveys important information about this architecture. Six filters were defined to characterize regional traits in copy number profiles, and the resulting Copy Aberration Regional Mapping Analysis (CARMA) algorithm was applied to tumors in four breast cancer cohorts ($n = 2919$). The derived motifs represent a layer of information that complements established molecular classifications of breast cancer. A score reflecting presence or absence of motifs provided a highly significant independent prognostic predictor. Results were consistent between cohorts. The nonsite-specific occurrence of the detected patterns suggests that CARMA captures underlying replication and repair defects and could have a future potential in treatment stratification.

[1] Department of Cancer Genetics, Institute for Cancer Research, Oslo University Hospital, Ullernchausseen 70 N-0310, Oslo, Norway. [2] Centre for Bioinformatics, Department of Informatics, University of Oslo, Gaustadalléen 23 B N-0373, Oslo, Norway. [3] Cancer Research UK, Cambridge Research Institute, Li Ka Shing Centre, Robinson Way, Cambridge CB2 0RE, UK. [4] Department of Mathematics, University of Oslo, Moltke Moes vei 35 N-0851, Oslo, Norway. [5] Institute of Basic Medical Sciences, Faculty of Medicine, University of Oslo, Domus Medica, Sognsvannsveien 9 N-0372, Oslo, Norway. [6] Centre for Molecular Medicine Norway, University of Oslo, Forskningsparken, Gaustadalléen 21 N-0349, Oslo, Norway. [8] Department of Pathology, Oslo University Hospital, POB 4953 Nydalen N-0424, Oslo, Norway. [9] Institute for Clinical Medicine, University of Oslo, Kirkeveien 166 N-0450, Oslo, Norway. [10] Department of Oncology, Oslo University Hospital, POB 4953 Nydalen N-0424 Oslo, Norway. [14] KG Jebsen Centre for B-cell malignancies, Institute for Clinical Medicine, University of Oslo, Ullernchausseen 70 N-0372, Oslo, Norway. [18] Big Data Institute, Li Ka Shing Centre for Health Information and Discovery, University of Oxford, Old Road Campus, Headington, Oxford OX3 7FZ, UK. [19] NIHR Biomedical Research Centre, Warneford Ln, Headington, Oxford OX3 7JX, UK. [20] The Francis Crick Institute, 1 Midland Road, London NW1 1AT, UK. [21]These authors contributed equally: Arne V. Pladsen, Gro Nilsen. [22]These authors jointly supervised this work: Hege G. Russnes, Ole Christian Lingjærde. *A list of authors and their affiliations appears at the end of the paper. ✉email: ole@ifi.uio.no

The allele-specific DNA copy number profile of a tumor is a window into its past history and its future evolutionary potential[1–3]. In general, we may consider a copy number profile as the accumulated result of a series of genomic events[4–7]. Specific DNA replication and repair errors may leave particular traces throughout the genome in the form of recurring local patterns, or motifs[8–11]. We hypothesized that such motifs represent a substantial proportion of the copy number variation in a tumor, and that they partly explain the high intertumor copy number heterogeneity frequently observed in cancer. We further hypothesized that the presence or absence of specific motifs is informative of a tumor's past and future evolutionary trajectory. Detailed characterization of such features would thus allow prediction of disease behavior and could potentially direct choice of treatment.

Here, we present an analysis of regional nonsite-specific motifs from allele-specific DNA copy number profiles in breast cancer. The core of this framework is the Copy Aberration Regional Mapping Analysis (CARMA) algorithm, which creates a compact representation of the aberration architecture. Conceptually, the algorithm represents copy number profiles as real-valued functions over the genomic domain and derives a small set of scores representing distinct regional features. The proposed method takes into account copy number amplitude, spatial distribution of copy number break points and allelic imbalance, and captures regional fluctuations in copy number, a signature feature of chromothripsis and chromoplexy. By generating a low-dimensional representation of the copy number data, the proposed algorithm also avoids the curse of dimensionality.

CARMA is related to multiple algorithms designed to detect specific copy number aberration patterns in tumors. The chromosomal instability index (CINdex)[12] and the genomic instability index (GII)[13] both quantify the total amount of genomic aberrations. Other algorithms have been proposed for detection of simplex and complex copy number events[9] and structural rearrangement patterns[14], for example the complex arm-wise aberration index (CAAI). An algorithm identifying the presence of multiple aberration patterns with application to ovarian cancer was recently proposed[11]. In addition, several methods have been proposed to identify copy number features recurring across tumors, such as GISTIC[15,16].

We applied CARMA to four breast cancer patient cohorts (METABRIC, Oslo2, Oslo-Val, and ICGC; see "Methods" for details). An integrated score was derived and shown to have superior prediction performance for breast cancer specific survival compared with other available clinical and molecular stratifications. The relation between copy number motifs and established driver gene based classifications of breast cancer was investigated. The analysis described in the paper is applicable to allele-specific copy number data from all types of cancer and any type of platform, including SNP arrays and high-throughput sequencing.

## Results

### Brief outline of the analysis approach.
CARMA is applicable to allele-specific copy number profiles from one or several tumors, obtained from SNP array analysis or DNA high-throughput sequencing. The algorithm extracts multiple local features which are accumulated across genomic regions by numerical integration to form six regional scores. These scores reflect the degree of amplification (AMP), deletion (DEL), complexity (STP and CRV), such as chromothripsis and chromoplexy, loss of heterozygosity (LOH) and allelic imbalance or asymmetry (ASM). More details and precise mathematical definitions are deferred to "Methods." The analysis pipeline is depicted in Fig. 1a–d. An

application of the algorithm to three breast tumor samples in the Oslo2 cohort and with chromosome arms as regions is shown in Fig. 1e. Specific regional features are discernible, illustrating how CARMA can be used to perform between-sample comparison of copy number features that are not locus specific.

### Relation to other methods.
CARMA was compared with two methods for detection of nonsite specific copy number aberrations in single samples: CAAI[9] and CINdex[12]. The CAAI algorithm identifies chromosome arms with complex rearrangements, while CINdex detects regional gains and losses. We also compared CARMA with GISTIC, a well-established method for detection of regions with significant copy number change across multiple samples[15,16]. Figure 2a shows circos plots of CARMA profiles for two selected samples in the METABRIC cohort, together with the results from GISTIC, CINdex, and CAAI.

As expected, CAAI correlates with the two CARMA complexity scores STP and CRV, but the relative sizes of STP and CRV provide additional detail (e.g., on chromosome 16 in the sample MB-0010). CINdex captures both gains and losses, but in the two selected samples it correlates stronger with DEL than with AMP. This is not unexpected, since the CINdex algorithm includes a relative weighting of gains and losses, while CARMA does not. The use of six distinct measure of copy number distortion in CARMA generally provides more detail than CINdex. For example, in a region with loss of one allele and gain of the other (i.e. a uniparental disomy), such as chromosome 22 in MB-0010, CARMA reports LOH and ASM, while CINdex reports no alteration (Fig. 2a). Observe also that the complex aberration on chromosome 11p in MB-0028 which is reported by CINdex is positive for all six CARMA scores including STP and CRV.

For GISTIC, regions of significant gain or loss were identified based on all METABRIC samples; a binary score is subsequently assigned to each sample in each such region based on the presence or absence of a loss or gain. Regions with significant loss or gain according to GISTIC partially overlap with DEL and AMP, respectively. Next, we investigated the distribution of CARMA scores within each region identified by GISTIC (Fig. 2b). A strong overlap is observed between GISTIC gain and high AMP score, and between GISTIC loss and high DEL score. In addition, there is considerable diversity in the CARMA spectrum within regions called as gains or losses according to GISTIC. For example, the relative contribution of LOH is highly variable across GISTIC loss regions. Similarly, the relative contribution of complex aberrations captured by STP and CRV varies across GISTIC gain regions.

### Molecular subgroups have distinct CARMA signatures.
We next considered the distribution of CARMA scores within established molecular stratifications of breast carcinomas (PAM50 and IntClust). PAM50[17,18] is an expression based classification system defining five distinct subgroups of breast tumors based on the correlation to a set of 50 genes. IntClust[1,19] identifies ten different subtypes based on the pattern of copy number aberrations exerting an effect on gene expression in cis. The distribution of CARMA scores within these classification systems were explored in four different breast cancer data sets of varying sample size ($n = 1943$, $n = 276$, $n = 165$, and $n = 553$). The percentage of tumors with scores exceeding a median threshold was plotted for all arm scores and for each PAM50 and IntClust subtype separately (Fig. 3a and Supplementary Figs. 1–4). The CARMA scores consistently reflect differences in the landscapes of genomic architecture in the different biological and clinical patient groups. This visual overview of aberration patterns highlights subtype specific features such as frequent allelic loss on

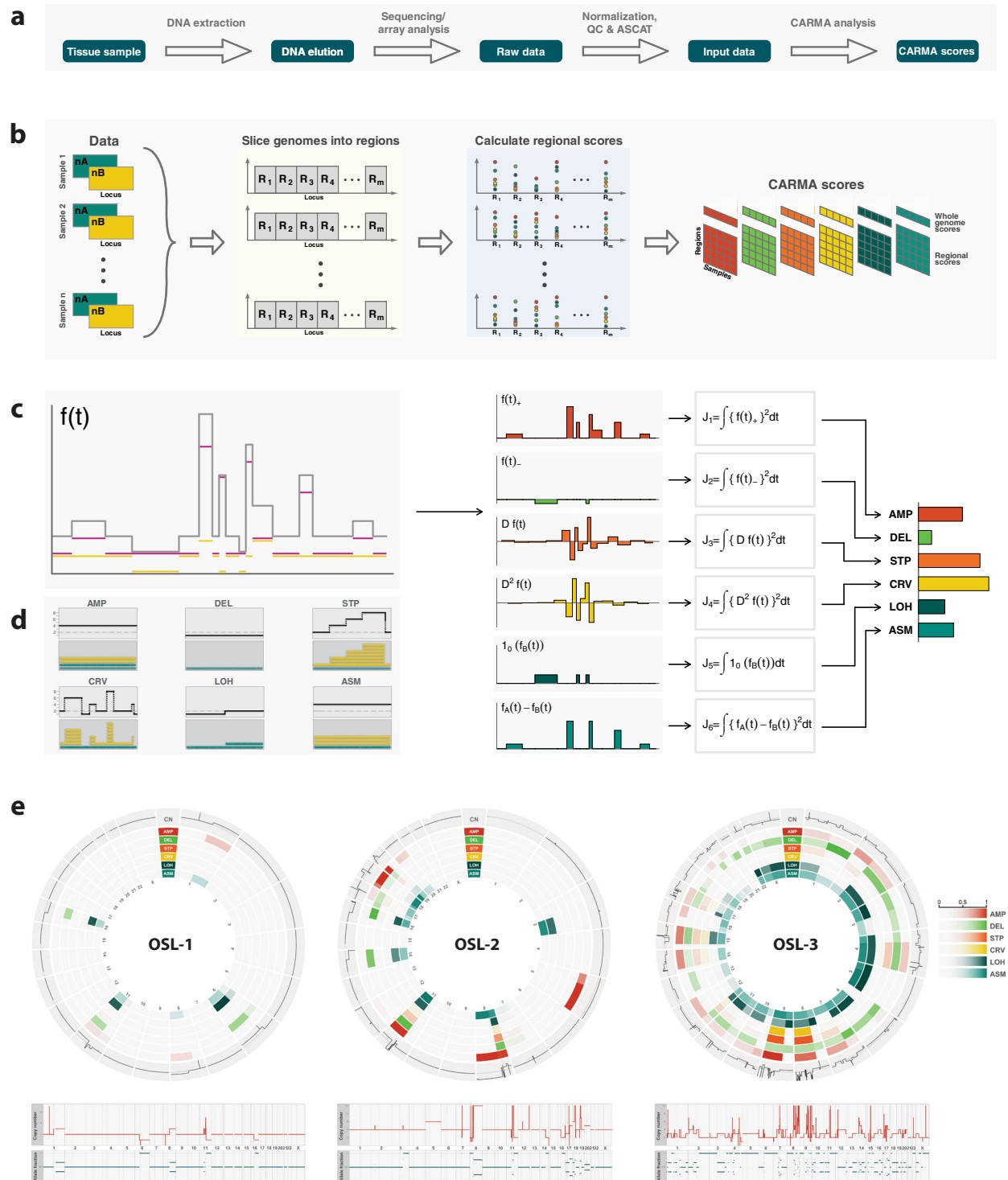

**Fig. 1 Outline of the CARMA algorithm. a** Complete analysis pipeline. **b** Steps included in the CARMA analysis. The input is one or more allele-specific copy number profiles. The algorithm extracts local features and accumulates these across genomic regions to form six regional scores. **c** Calculation of CARMA scores within a specified region. **d** Prototype patterns captured by each of the six CARMA scores. **e** An application of the algorithm to three breast tumor samples in the Oslo2 cohort. Lower panel: total copy number and allele fraction as a function of genomic locus. Upper panel: circos plots of regional (arm-wise) CARMA scores.

17p and frequent gain and high complexity on 17q in IntClust1; gain on 1q, frequent asymmetric gain and complex aberrations on 11q and allelic loss on 16q in IntClust2; etc. The signatures of regional CARMA scores within the PAM50 subtypes highlight known features, including whole arm 1q gain/16q loss in luminal A tumors, the more complex copy number aberrations in luminal B tumors, the 17q alterations dominating Her2-enriched tumors, and the global instability of basal-like tumors. Three-dimensional scatter plots of CARMA scores were plotted for all tumors in the Oslo2 cohort ($n = 276$) and METABRIC cohort ($n = 1943$) (see Fig. 3b). Trend curves and subtype centroids both demonstrate high degree of consistency between the two cohorts.

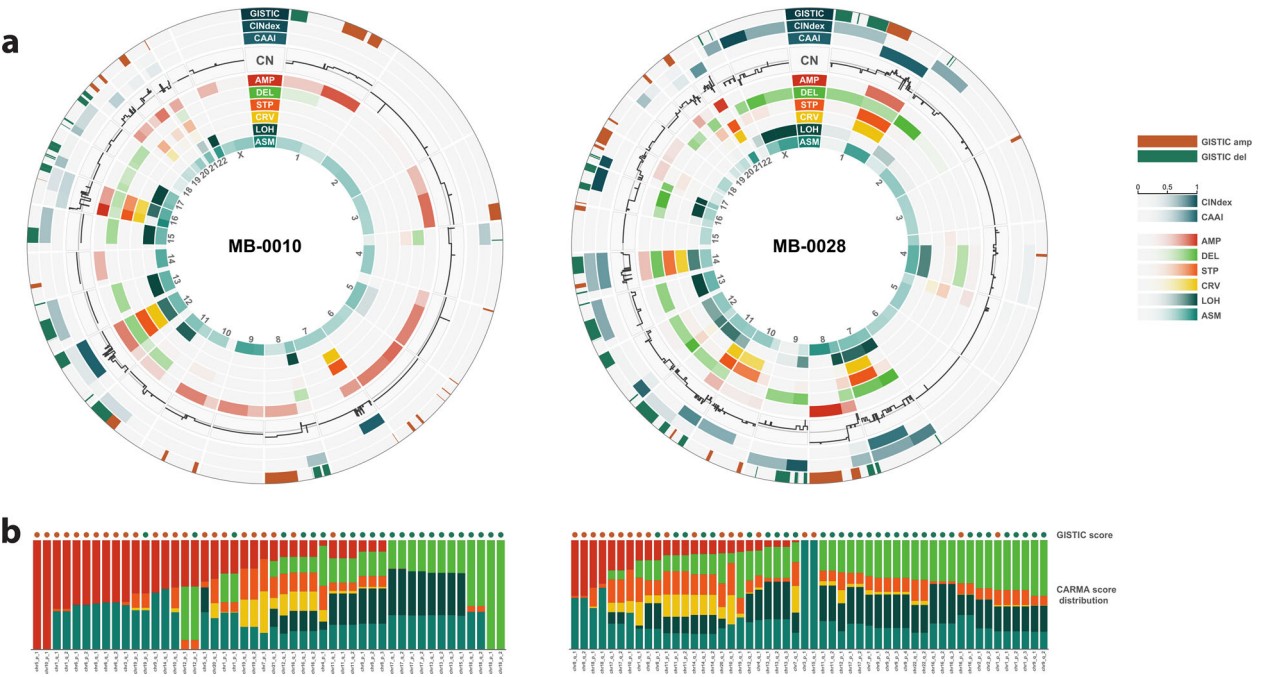

**Fig. 2 Relation to other methods. a** Circos plots of arm-wise CARMA scores for two selected samples in the METABRIC cohort, together with results from GISTIC, CINdex, and CAAI. Only focal aberrations are shown for GISTIC, and scores indicate whether a significant region according to GISTIC is aberrant in this sample. The color indicates the direction of change (loss or gain). For CINdex and CAAI, continuous scores are shown. **b** Relative distribution of CARMA scores within GISTIC regions. Each bar corresponds to a region which is found to be significant across all METABRIC samples according to GISTIC, and which is called as a gain or loss in the given sample. The colors in each bar represents the relative contributions of the six CARMA scores in that region, found by dividing the CARMA scores in a region by their sum. Regions are ordered according to decreasing contribution of AMP and then on increasing contribution of DEL.

**Predicting survival from regional scores**. To assess the association between disease-specific survival (DSS) and genome-wide CARMA scores, a univariate Cox proportional hazards regression model was fitted with each score as a covariate (see Supplementary Table 1). For this purpose, we used the largest cohort (METABRIC set). All scores were associated with survival ($P < 10^{-6}$; Score test) and the strongest associations were found for the scores STP and CRV ($P < 10^{-18}$; Score test).

We next split the METABRIC cohort into a discovery cohort ($n = 1295$) and a test cohort ($n = 648$). We fitted a multivariate Cox regression model to DSS and progression-free survival (PFS) data in the discovery cohort based on the six predictors. The predictors were defined by taking an unweighted mean across all the regional (arm-wise) CARMA scores (Fig. 3c). The fitted model was next applied to the test set, producing a single unweighted prognostic value per patient. Thresholds corresponding to the 1/3 and 2/3 percentile were applied to classify samples into groups of low, intermediate, and high risk, with numerical values ranging from 1 to 3. This final score was termed the CARMA Prognostic Index (CPI). An alternative prognostic index was defined using the 252 arm-wise CARMA scores directly as predictors and fitting a Cox regression model with Lasso penalty to the training set. Coefficients derived from the analysis (Supplementary Fig. 5) were used as weights to calculate a weighted prognostic index termed CPI$_{weighted}$.

To compare the efficacy of CPI and CPI$_{weighted}$ to established clinically and biologically relevant parameters, we fitted a univariate Cox regression model in the METABRIC test set using the prognostic indices and the clinical parameters as covariates (Table 1 and Supplementary Tables 2–3). The $P$ value for CPI from the analysis was lower than for any of the other clinical parameters when looking at both DSS ($P = 1.9 \times 10^{-13}$; Score test) and PFS ($P = 5.7 \times 10^{-13}$; Score test), and also

performed better than CPI$_{weighted}$. However, CPI$_{weighted}$ did remain strongly significant in the analysis for both DSS ($P = 5.2 \times 10^{-10}$; Score test) and PFS ($P = 3.7 \times 10^{-7}$; Score test) presenting $P$ values lower than many of the other established parameters. Hazard ratios for CPI and other clinical variables from univariate Cox regression analysis are shown in Fig. 3d.

Cox regression modeling was also performed to assess the effect of the prognostic indices with adjustments for other variables (see Table 1 and Supplementary Tables 2–3). CPI consistently showed smaller $P$ values than all other clinical variables. Also CPI$_{weighted}$ remained significant when adjusting for other variables (Supplementary Tables 2–3). Hazard ratios from multivariate Cox regression models where the effect of CPI is adjusted for the effect of clinical variables are shown in Fig. 3f.

CPI was next used to stratify patients into low, intermediate, and high-risk groups as described above in the three validation cohorts with survival data available (METABRIC test set, OsloVal, and ICGC). A logrank test was performed for the three groups in each data set (Fig. 3e). $P$ values were significant when considering both DSS ($P < 10^{-12}$ in METABRIC test, $P < 10^{-4}$ in OsloVal, and $P = 0.003$ in ICGC) and PFS ($P < 10^{-12}$ in METABRIC test; PFS data were not available for OsloVal or ICGC).

Finally, the unweighted continuous prognostic score that was used to obtain the CPI, was utilized to calculate a Harrell's C score in the METABRIC test set. The C scores obtained from the analysis were 0.65 (95% CI: 0.62–0.69) and 0.64 (95% CI: 0.61–0.68) based on DSS and PFS, respectively.

## Discussion

Structural DNA distortions are a result of deregulated DNA repair and maintenance, and mutagenic processes operating in the cells. The conventional focus in studies of DNA copy number

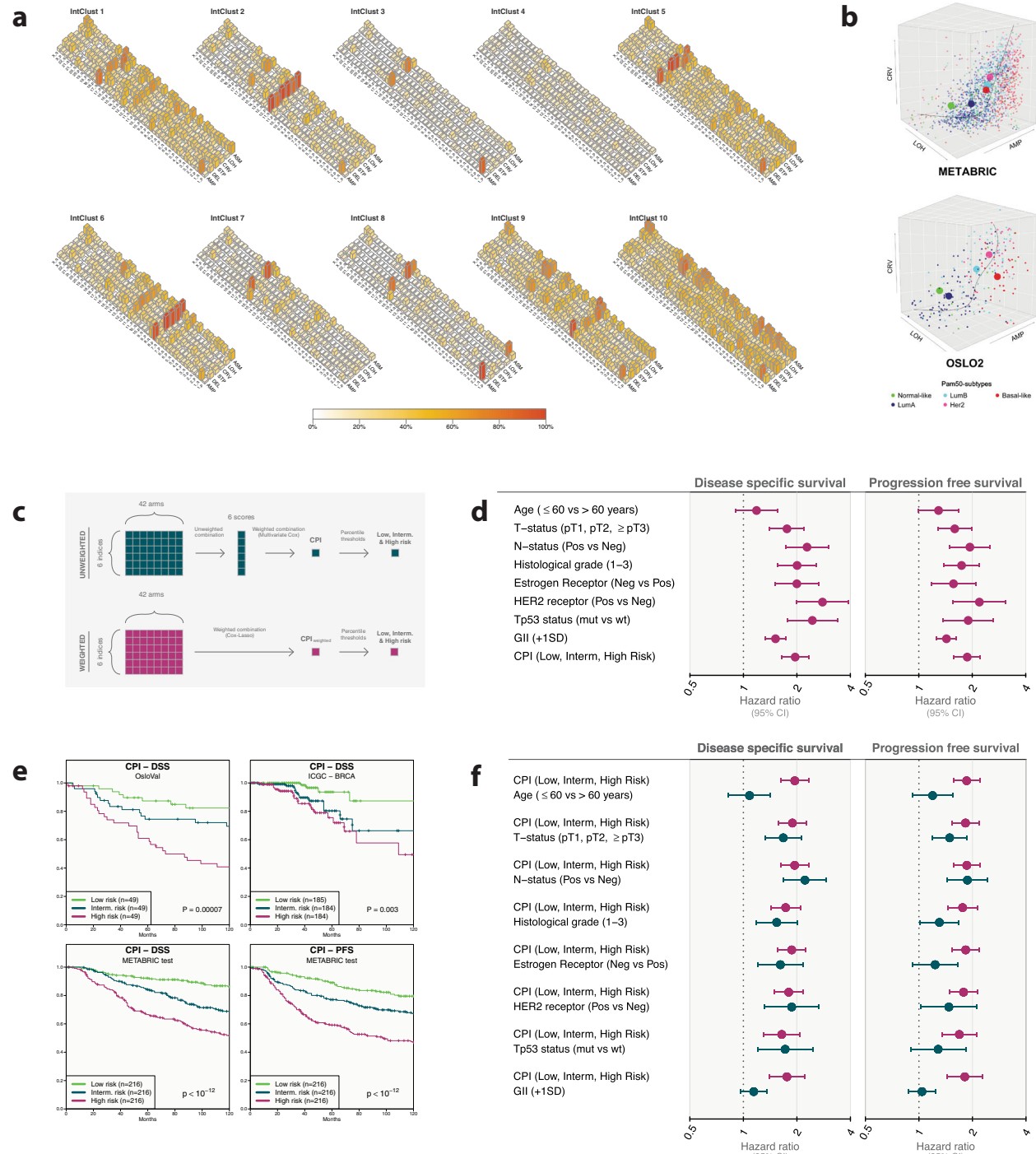

**Fig. 3 Stratification and outcome prediction with CARMA. a** CARMA score distribution in METABRIC within the IntClust subtypes defined in Curtis et al.[1]. The height of each bar represents the proportion of samples in the subgroup with arm score above the median, calculated across all arms within each CARMA score and ignoring zeros. **b** Three-dimensional scatter plots of tumors using three of the CARMA scores designed to detect three major categories of copy number aberration patterns in tumors (amplifications AMP, allelic loss LOH, complex rearrangements CRV). Colors indicate PAM50 subtype (see legend at bottom) and large spheres show subtype centroids. Upper panel: Oslo2 (*n* = 276); Lower panel: METABRIC (*n* = 1943). **c** Flow chart depicting the construction of prognostic indices from the arm-wise CARMA scores, using the METABRIC discovery cohort. Upper panel: construction of CPI. Arm-wise scores are collapsed using an unweighted average, and the resulting genome-wide scores are combined by multivariate Cox regression. Thresholds corresponding to the 1/3 and 2/3 percentiles were applied to classify samples into groups of low, intermediate and high risk. Lower panel: construction of CPI$_{weighted}$. Arm-wise scores are combined by cross-validated multivariate Cox-Lasso regression, resulting in one genome-wide score. Thresholds corresponding to the 1/3 and 2/3 percentiles were applied as above to classify samples into three risk groups. **d** Hazard ratios and 95% confidence intervals (CI) for clinical variables, CPI scores, and the genomic instability index (GII). Shown are unadjusted estimates for disease-specific survival (DSS) and progression-free survival (PFS). **e** Survival prediction using CPI stratified into low, intermediate, and high risk. Kaplan–Meier plots of DSS for the three risk groups in the METABRIC test set, OsloVal set, and ICGC set as well as for PFS within the METABRIC test set. **f** Hazard ratios and 95% CI for clinical variables, CPI, and GII. Shown are adjusted estimates for DSS and PFS.

**Table 1 Prognostic value of CPI and other variables.**

| Type | Covariate | Disease-specific survival | | Progression-free survival | |
|------|-----------|---------|------|---------|------|
| | | *P* value | HR | *P* value | HR |
| Unadjusted | Age (≤60 vs >60 years) | 2.2e−01 | 1.19 | 5.3e−02 | 1.29 |
| Unadjusted | T-status (pT1, pT2, ≥pT3) | 9.2e−07 | 1.76 | 2.9e−05 | 1.59 |
| Unadjusted | N-status (positive vs negative) | 5.5e−09 | 2.28 | 8.2e−07 | 1.94 |
| Unadjusted | Histological grade (1–3) | 5.9e−08 | 2.01 | 3.2e−06 | 1.74 |
| Unadjusted | Estrogen receptor (negative vs positive) | 2.0e−06 | 2.00 | 2.3e−03 | 1.57 |
| Unadjusted | HER2 receptor (positive vs negative) | 1.6e−09 | 2.79 | 7.0e−06 | 2.19 |
| Unadjusted | Tp53 status (mutated vs wildtype) | 7.2e−08 | 2.44 | 1.0e−04 | 1.90 |
| Unadjusted | GII (+1 SD) | 1.1e−09 | 1.52 | 3.6e−08 | 1.43 |
| Unadjusted | CPI (low, intermediate, high risk) | 1.9e−13 | 1.96 | 5.7e−13 | 1.87 |
| Adjusted | CPI (low, intermediate, high risk) | 3.4e−13 | 1.95 | 1.5e−12 | 1.85 |
| | Age (≤60 vs >60 years) | 5.6e−01 | 1.08 | 1.8e−01 | 1.19 |
| Adjusted | CPI (low, intermediate, high risk) | 5.5e−12 | 1.89 | 9.0e−12 | 1.83 |
| | T-status (pT1, pT2, ≥pT3) | 1.1e−05 | 1.68 | 4.6e−04 | 1.49 |
| Adjusted | CPI (low, intermediate, high risk) | 5.6e−13 | 1.94 | 1.9e−12 | 1.86 |
| | N-status (positive vs negative) | 1.8e−08 | 2.22 | 3.1e−06 | 1.87 |
| Adjusted | CPI (low, intermediate, high risk) | 2.2e−08 | 1.73 | 4.2e−09 | 1.76 |
| | Histological grade (1–3) | 1.5e−03 | 1.54 | 3.7e−02 | 1.30 |
| Adjusted | CPI (low, intermediate, high risk) | 2.0e−11 | 1.87 | 1.5e−11 | 1.83 |
| | Estrogen receptor (negative vs positive) | 1.2e−03 | 1.62 | 1.6e−01 | 1.23 |
| Adjusted | CPI (low, intermediate, high risk) | 6.7e−10 | 1.80 | 1.9e−10 | 1.78 |
| | HER2 receptor (positive vs negative) | 4.6e−04 | 1.87 | 3.4e−02 | 1.47 |
| Adjusted | CPI (low, intermediate, high risk) | 3.1e−05 | 1.64 | 5.6e−06 | 1.69 |
| | Tp53 status (mutated vs wildtype) | 3.1e−03 | 1.72 | 1.7e−01 | 1.28 |
| Adjusted | CPI (low, intermediate, high risk) | 1.2e−06 | 1.76 | 4.5e−07 | 1.81 |
| | GII (+1 SD) | 1.3e−01 | 1.14 | 6.6e−01 | 1.04 |

The first part of the table shows results from univariate Cox regression analyses to assess the association between survival and clinical variables, GII, and CPI. Results are shown for both DSS and PFS and for the METABRIC test cohort ($n = 648$). The second part of the table shows results from multivariate Cox regression analyses on the sample samples to assess the association between CPI and survival, with adjustment for the effect of other variables.
*HR* hazard ratio.

alterations in tumors is the identification of recurrently deleted and amplified genes which may define key driver events in carcinogenesis or potential targets for treatment. We and others have previously shown that in addition to this gene centered or locus centered approach, the structural changes provide important information for classification and survival prediction[8,9,20]. The methodology presented in this study complements gene specific analyses by providing a systematic framework to characterize the information embedded in the copy number profile of a tumor. CARMA determines the presence and relative contributions of six distinct copy number features in genomic regions and in the genome as a whole. By focusing on pervasive patterns or motifs in the genome rather than locus specific events, the algorithm captures footprints of past and ongoing segmental DNA alterations. Known drivers of such alterations are DNA replication and repair errors[8–11].

In this study, we used CARMA to assign scores to individual chromosome arms and to the whole genome. The CARMA algorithm is not bound to any particular genomic resolution though, and the tool supports assignment of individual scores to whole genomes, chromosomes, chromosome arms, or genomic bins of any desired width. For a given genomic resolution, scores for individual genes can also be obtained by inheritance of the respective regional score. Irrespective of the selection of regions on which to assign scores, the fact that regions are identical across tumors allows CARMA scores to be used directly as features in clustering, regression, and classification. Normally, the number of features will also be quite small, thus substantially reducing statistical problems related to high dimensionality.

CARMA reveals a rich spectrum of different copy number motifs across samples and also between regions within an individual sample. By combining six different measures of copy number aberration, it provides a more detailed picture of genomic architecture than GII, CAAI, and CINdex. CARMA and GISTIC represent complementary tools with different aims. Combining CARMA with GISTIC offers the possibility of providing a detailed picture of the aberration spectrum restricted to regions that are significantly altered across many samples.

Molecular taxonomy of breast cancer based on gene expression has proved important for the biological understanding of the disease[17]. IntClust[1] is a more recent driver-based classification of breast cancer and has been shown to also reflect degree of chemosensitivity[21]. The CARMA scores revealed distinct aberration signatures for the ten IntClust groups, suggesting that the copy number motifs reflect a driver-based classification of tumors. As seen from the Manhattan plots, the expression signatures defining the IntClust subtypes are to a large degree correlated to focal copy number aberrations, representing driver alterations in these subtypes. The copy number aberrations in these driver regions also exhibit differences in their pattern. This is for instance illustrated by the different types of copy number gains found on the 1q arm in the IntClust 8 subtype, as compared with the gains found on the 11q arm in the IntClust2 group. The first type of gain represents noncomplex low-amplicon whole arm translocations captured by the AMP and ASM scores, while the latter represents more complex rearrangements with high-amplicon gains[22] captured by all of the CARMA scores. Even though both of the observed patterns represent copy number gains, the underlying mechanisms causing these patterns are fundamentally different. The CARMA scores manage to capture these nuances, illustrating the potential of the method to discriminate between a richer set of aberrational patterns. The plot also gives an indication of the global background variation from copy number aberrations, maybe most apparent in the IntClust ten subtypes.

Interestingly, the degree to which the different subtypes are affected by this background variation seems to correlate well with the fraction of *TP53* mutations observed within each subtype[23]. This again supports the notion that copy number motifs reflect underlying biological traits.

In order to assess the ability of the method to predict breast cancer specific survival, a univariate Cox regression model was fitted to genome-wide CARMA scores in the METABRIC cohort. All genome-wide scores showed a strong and significant association to survival. As a first step this supports the assumption that each of the selected scores are informative and thus qualifies for use in further survival analyses. The scores were combined to produce the unweighted and weighted prognostic indices CPI and $CPI_{weighted}$. When CPI and $CPI_{weighted}$ were compared with established clinical parameters through Cox regression analyses, CPI consistently outperformed all other variables in terms of the level of significance. The multivariate Cox analyses established that CPI is a strong independent predictor of survival in breast cancer. The results might point towards a role of specific aberration motifs, proceeding from specific types of genomic instability, as determinants of malignancy potential in a tumor. The fact that CPI outperformed GII in the above analyses supports the idea that additional information is added through multifaceted measurements of copy number aberrations.

The observation that CPI produced better prognostic predictions than $CPI_{weighted}$ mightstem from the somewhat strict variable selection exerted by the Lasso regression model. The Lasso model excludes arm-specific scores that individually do not contribute strongly to the survival prediction. Aggregated, however, these arm-specific scores might confer additional prognostic information. CPI, which is based on combining all arm scores in an unweighted manner, is not subject to the same kind of selection bias. The fact that this more inclusive approach performed better in our analyses suggests that all parts of the genome copy number aberration profile contribute to the real signal when assessing survival. This supports the notion that our method captures omnipresent background variation caused by underlying DNA disruptions.

In the future it would be of high interest to apply the methodology to different cancer types to compare aberration patterns across tumors at different sites, for example using The Cancer Genome Atlas Pan-Cancer data set[24]. Translocation of genomic material is not captured by any array-based DNA analysis, and data from high-throughput sequencing would be required to fully characterize genomic architecture. The complex patterns described in this manuscript are likely to reflect specific mutational processes that could be further elucidated in future studies, linking CARMA with sequencing data. Finally, ASCAT has recently been implemented for whole genome sequencing data[25], and it would be interesting to apply our methodology directly to the allele-specific copy number profiles extracted from such data.

Several extensions of the current analyses are possible. One could for example in- crease the genomic resolution by partitioning the genome into a fairly large number of equal-sized regions (say 1000), and then assign separate scores to each of these. At some point, however, the regions may become too small to meaningfully assign scores, most notably for the indices reflecting complex rearrangements (STP and CRV). Another possible extension would be to consider regions harboring genes involved in specific processes or pathways, thus directly linking CARMA scores to biological function.

## Methods

**Deriving allele-specific copy number profiles**. Affymetrix CEL files were preprocessed using the PennCNV libraries for Affymetrix data[26] that includes quantile normalization, signal extraction, and summarization. All samples were normalized

to a collection of around 5000 normal samples from the HapMap project[27], the 1000 genome project[28], and the Wellcome Trust Case Control Consortium[29]. The resulting LogR and BAF (B allele frequency) values were segmented with the piecewise constant fitting algorithm[30] and processed with the ASCAT algorithm (version 2.3)[31] after adjusting LogR for GC binding artifacts[32]. ASCAT infers an allele-specific copy number profile of a tumor after correction for tumor ploidy and tumor cell fraction, and is based on allele-specific segmentation of normalized raw data[30] with penalty parameter ($\gamma$) set to 50. The profile reflects the copy number state at $m$ genomic loci for which two alleles are present in the germline in the general population, and can be represented as a sequence of pairs ($n_{Ai}$, $n_{Bi}$) ($i = 1$, ..., $m$), where $n_{Ai}$ and $n_{Bi}$ denote the number of copies of each of two alleles (here called A and B) being present in the tumor genome at the $i$th locus. Pairs are ordered according to location, and since the labels A and B are arbitrary, we may assume that $n_{Ai} \geq n_{Bi}$.

**Calculating regional instability scores**. We characterize the allele-specific copy number in a small genomic neighborhood on a chromosome arm by six features: degree of alteration in negative direction, degree of alteration in positive direction, degree of change, degree of oscillation, extent of LOH, and extent of allelic imbalance (see Fig. 1c). Sliding the genomic region along the chromosome arm from one end to the other, we may regard each feature as a function of genomic position. Specifically, suppose we have measured allele-specific copy numbers ($n_{Ai}$, $n_{Bi}$) at genomic loci $L_i$, $i = 1,...,m$. We can represent this as a pair of piecewise constant functions ($f_A$, $f_B$) defined on the unit interval $R = [0, 1]$. The interpretation of this is that each position $L_i$ is mapped to a value $t_i$ in the unit interval $R = [0, 1]$, and such that $L_1 < \cdots < L_m$ will be represented by points $t_1 < \cdots < t_m$ in $R$. We thus have a one-to-one correspondence between $t \in [0, 1]$ and genomic loci $L(t)$, and if $L_k$ is the measurement locus closest to $L(t)$, then $f_A(t) = n_{Ak}$ and $f_B(t) = n_{Bk}$. We assume that $f_B(t) \leq f_A(t)$ for all $t \in R$, i.e., B is the minor allele when allelic imbalance is present. The median centered total copy number in locus $t$ is $f(t) = f_A(t) + f_B(t) - m$, where $m$ is the least number in Range($f$) that satisfies $\mu(f^{-1}((-\infty, m])) \geq 1/2$, where $\mu$ is the Lebesgue measure. Informally, this means that $m$ is chosen as the observed copy number with the property that half the genome has a total copy number less than or equal to $m$. We define the change in total copy number as the derivative $Df(t)$ of the first order spline interpolation to the center points of segments in $f$, i.e. $Df(t)$ is the slope of the line segment connecting the pair of segment centers immediately to the left and right of position $t$. Note that $Df$ is also a piecewise constant function. We define the oscillation in total copy number as $D^2f(t) = D(Df(t))$, which is also a piecewise constant function. This process can in principle be repeated to define higher order properties of $f$ such as $D^3f(t) = D(D^2f(t))$; however, in practice further levels add little additional information.

Regional instability scores are next defined by integrating the above local scores over the desired region (e.g., over a chromosome arm). To assess the degree of positive or negative deviation within a region, we define two scores:

$$J_1 = \int_R \{f(t)_+\}^2 dt \text{ and } J_2 = \int_R \{f(t)_-\}^2 dt,$$

where $z_+ = z$ if $z > 0$ and $z_+ = 0$ otherwise, and $z_- = z$ if $z < 0$ and $z_- = 0$ otherwise. For example, in a region with total copy number equal to the median, we have $J_1 = J_2 = 0$, while in a region with some gains and no losses relative to the median, we have $J_1 > 0$ and $J_2 = 0$. The regional degree of change and oscillation in copy number are captured by the following two scores:

$$J_3 = \int_R \{Df(t)\}^2 dt \text{ and } J_4 = \int_R \{D^2f(t)\}^2 dt.$$

In a region with constant total copy number, we have $J_3 = J_4 = 0$. In a region with gradually increasing (or decreasing) copy number, $J_3 > 0$ while $J_4$ is close to zero, and in a region with fluctuations between smaller and larger copy numbers we have $J_3 > 0$ and $J_4 > 0$. LOH and allelic asymmetry are captured by the last two scores:

$$J_5 = \int_R \{1_0(f_B(t))\} dt \text{ and } J_6 = \int_R (f_A(t) - f_B(t))^2 dt,$$

where $1_0(z) = 1$ if $z = 0$ and $1_0(z) = 0$ otherwise. In a region with only one allele present we have $J_5 > 0$ and the magnitude of the score reflects the proportion of the region with LOH. In a region with allelic imbalance, we have $J_6 > 0$. Further computational details can be found in Supplementary Materials.

**Calculating CARMA scores in sex chromosomes**. The top level function in the accompanying software does not currently support calculation of CARMA scores for the Y chromosome. It is still possible to obtain such scores by use of the included low level function for calculating scores on a single chromosome. Calculation of CARMA scores for the X chromosome is supported, but it requires information about the gender for correct calculation of AMP and DEL.

**Statistics and reproducibility**. Three-dimensional scatter plots: Subtype centroids were calculated by averaging over all the three-dimensional vectors representing

samples from a particular PAM50 subtype. Trend curves are principal curves[33] and were calculated with the R package `princurve` using default parameter values.

Survival analysis: To assess the association between survival (DSS or PFS) and CPI risk groups, a longrank test was applied, and survival estimates were found using the Kaplan–Meier estimator. The functions `survdiff` and `survfit` in the R package `survival` were used for this purpose. All other associations between survival and covariates were assessed using univariate or multivariate Cox regression, as appropriate. A score test was applied to test the significance of individual covariates in the Cox models. Models were fitted by maximization of the Cox partial likelihood, with the exception of the model containing all the 252 arm-wise CARMA case a Cox partial likelihood with an $L_1$ (lasso) penalty[34] was applied. The lasso is a regularization method that shrinks regression coefficients towards zero by enforcing an upper bound on the $L_1$-norm of the coefficients (i.e. $\sum_{j=1}^{p} |\beta_j| \le \lambda$) in the maximization of the partial log likelihood.

The amount of shrinkage is determined by a tuning parameter $\lambda$. Leave-one-out cross-validation was used to determine the value of $\lambda$. Cox regression with a Lasso penalty was performed using the functions `cv.glmnet` and `glmnet` in the R package `glmnet`[35,36]. All other Cox regressions were performed using the function `coxph` in the R package `survival`.

Assessment of risk-score model: The goodness of fit of the continuous CPI risk score was determined using Harrell's C score. For every pair of observations it is determined if the pair is concordant (lowest risk pairs with longest survival), discordant (lowest risk pairs with shortest survival) or cannot be determined due to censoring. Harrell's C score is then the ratio between the number of concordant pairs and the number of concordant/discordant pairs.

The weighted prognostic index (CPI$_{weighted}$) was calculated as $\mathrm{CPI}_{weighted} = x_i^T \hat{\beta}$, where $x_i$ represents the CARMA arm scores for patient $i$ in the validation data set and $\hat{\beta}$ are the estimated coefficients in the survival prediction models found for the discovery set.

**Materials**. The data material in this study was obtained from four patient cohorts: METABRIC ($n = 1943$), Oslo2 ($n = 276$), OsloVal ($n = 165$), and ICGC ($n = 553$). Only female patients were included. The distribution of clinical parameters within each of the data sets can be found in Supplementary Tables 4–5. The METABRIC cohort was randomly split into a 2:1 ratio into a discovery set ($n = 1295$) and a test set ($n = 648$) for the purpose of model validation. For detailed information regarding which samples belong to the train and test cohort, please contact the authors. For more details about the four cohorts, see Supplementary Material and Methods. Survival data were not available for the Oslo2 cohort.

**Reporting summary**. Further information on research design is available in the Nature Research Reporting Summary linked to this article.

## Data availability

Genomic copy number and gene expression information as well as clinical data for the OsloVal cohort have been described previously[37] and are available at the Synapse platform, https://doi.org/10.7303/syn1688370. Gene expression information for the Oslo2 cohort has been described previously[38,39] and is available at Gene Expression Omnibus, DOI: GSE81002. The SNP 6.0 copy number data from the Oslo2 cohort are available upon request. Molecular-subtype information and segmented copy number profiles for the OsloVal and Oslo2 cohort are available from the corresponding author on reasonable request. Genomic copy number, gene expression and molecular-subtype information for the METABRIC cohort have been described previously[1] and are available at the European Genome Phenome Archive, DOI: EGAS00000000083, while clinical data are available from[19]. Gene expression data, segmented copy number profiles and clinical information for the ICGC breast cancer cohort have been described previously[40] and are available from the Supplementary Tables in that publication. Raw data are available at the European Genome Phenome Archive under the overarching accession number EGAS00001001178.

## Code availability

Software with detailed instructions and test data is available as an R package at the web site http://heim.ifi.uio.no/bioinf/Projects/. The software is open source and may be used according to the MIT license.

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

### Acknowledgements

The authors thank all the women who have contributed to this study by donating tumor tissue and blood. We thank Hans Kristian Moen Vollan for vital input and support in the development of an early version of the CARMA algorithm. We thank Sandra Jernstrøm for her assistance in preparing the gene expression data on Oslo2, Einar Rødland for his assistance in normalization of the gene expression data on Oslo2, and Phoung Vu, Veronica Skarpeteig, Inger Riise Bergheim, and Anja Valen for the *TP53* sequencing of Oslo2. David Wedge is supported by the Li Ka Shing Foundation and National Institute for Health Research Oxford Biomedical Research Centre. Peter Van Loo is supported by the Francis Crick Institute, which receives its core funding from Cancer Research UK (FC001202), the UK Medical Research Council (FC001202), and the Wellcome Trust (FC001202). Peter Van Loo is a Winton Group Leader in recognition of the Winton Charitable Foundation's support towards the establishment of The Francis Crick Institute.

### Author contributions

A.V.P., G.N., and O.C.L. performed the statistical and bioinformatical analyses, with contributions from O.M.R., M.R.A., Ø.B., K.L., V.V., A.F., A.M., O.E., D.C.W., P.V.L., and H.G.R. V.V., and A.F. performed the IntClust subtyping in the Oslo2 cohort. A.V.P., G.N., H.G.R., and O.C.L. developed the CARMA method, with contributions from M.R.A., O.E., V.K., and C.C. A.L., V.K., and A.L.B.D. performed and planned laboratory experiments. OSBREAC and A.L.B.D. provided patient materials. A.V.P., G.N., H.G.R., and O.C.L. conceived the study and wrote the manuscript. All authors performed critical revision of the manuscript and have read and accepted the final version.

### Competing interests

The authors declare that they have no competing interests.

### Additional information

## OSBREAC

Tone F. Bathen[7], Elin Borgen[8], Anne-Lise Børresen-Dale[1,9], Olav Engebråten[9,10], Britt Fritzman[11], Øystein Garred[8], Jürgen Geisler[12], Gry Aarum Geitvik[1], Solveig Hofvind[13], Vessela Kristensen[1], Rolf Kåresen[9], Anita Langerød[1], Ole Christian Lingjærde[1,2,14], Gunhild Mari Mælandsmo[15], Bjørn Naume[9,10], Hege G. Russnes[1,8], Kristine Kleivi Sahlberg[16], Torill Sauer[12], Helle Kristine Skjerven[16], Ellen Schlichting[17] & Therese Sørlie[1]

[7]Norwegian University of Science and Technology, N-7491 Trondheim, Norway. [11]Østfold Hospital, POB 300 N-1714, Grålum, Norway. [12]Akershus University Hospital, Sykehusveien 25, Lørenskog, Norway. [13]Cancer Registry of Norway, Ullernchausseen 64 N-0379, Oslo, Norway. [15]Department of Tumor Biology, Institute for Cancer Research, Oslo University Hospital, Ullernchausseen 70 N-0310, Oslo, Norway. [16]Vestre Viken Hospital Trust, POB 800 N-3004, Drammen, Norway. [17]Section for Breast and Endocrine Surgery, Division of Surgery, Cancer and Transplantation Medicine, Oslo University Hospital, N-0424 Oslo, Norway.

