## [Peer Review File · Communications Biology]

Reviewers' comments:

Reviewer #1 (Remarks to the Author):

The study develops a non-site specific algorithm to characterize the copy number variation of a genome and utilize it to predict survival. Since there are many non-site specific algorithm available. The author needs to compare their algorithm with the existing non-site specific or site specific algorithm to demonstrate better utility. In addition, other clinical and pathologically parameters should be included in the analysis, not just survival.

Reviewer #2 (Remarks to the Author):

The authors have developed an algorithm named Copy Aberration Regional Mapping Analysis (CARMA) to identify tumour copy number profile based motifs. Then they have proposed to use these motifs to develop a personalized score based on presence/absence of them, which can be used as an independent prognostic predictor for breast cancer. They have compare the CARMA signatures against breast cancer subtypes derived based on PAM50 genes and InClust.

Overall, I find the manuscript is well-written and include a good explanation of the method. If CARMA can be used in other cancer types easily and with a good documentation, I think the method will be very useful in many cancer types.

Major comments:

- 1) The software was not available for testing. It is recommended to have the code and test data set available for testing purposes. So, I couldn't run the programme.
- 2) The method is mainly devised around calculating six different types of scores (AMP, DEL, etc...). These abbreviations are first given in Figure 1. However, these abbreviations weren't introduced earlier. So, it is difficult to follow the figure and connect the figure with figure references under section 2.1. Please provide the description of these abbreviations before their first appearances.
- 3) Harrell's C scores are specified to assess the predictive power on survival data. It is recommended to have the confidence interval of these values to have a better understanding of the performance.
- 4) I couldn't find the reasoning behind the use of arm level scores rather than gene level or chromosome level CARMA scores. Please provide these details for improve the readability.
- 5) Did you only select female samples from the normal samples for the signal normalization? If not, how did you normalize for X chromosome? It is good to have these details in the methods section.
- 6) Can CARMA be applied to other cancer types? If yes, please provide some details on this. For example, handling chromosome X and Y scores.
- 7) How CARMA performance comparing against CINdex?
- 8) What are the differences between using CARMA to identify CN patterns compared to GISTIC? How CARMA gives better understanding of the CN patters?

Minor comments:

- 1) 'inn' should change to 'in' on page 5 last paragraph.
- 2) What is 't' in page 6? It is not clear in main text.

Reviewer #3 (Remarks to the Author):

The "DNA copy number motifs are strong and independent predictors of survival in breast cancer" paper

hypothesized that short-range copy number features present throughout the genome mirror underlying instability

processes. Thus, it uses the copy number features to predict breast cancer.

Here, I have the following questions.

First, could author provide a detailed flow chart to describe the CARMA algorithm?

Second, why the author did not consider the novel single cell sequencing or three-dimensional genome methods

to predict breast cancer?

Third, Authors only use three data sets to demonstrate their exploration and validate the efficiency of the

algorithm, which is not enough for a nature affiliated journal.

Thank you

Rebuttal Letter

Detailed responses to the reviewer comments are given below. Referee comments are in black, author response in blue, and actions taken are in blue italic. Two identical versions of the revised paper are submitted, the only difference being that changes have been highlighted in red in one of them.

Please see responses below

Reviewer # 1: The study develops a non-site specific algorithm to characterize the copy number variation of a genome and utilize it to predict survival. Since there are many non-site specific algorithm available. The author needs to compare their algorithm with the existing non-site specific or site specific algorithm to demonstrate better utility

We agree that it would be useful to compare our algorithm with existing methods. We have discussed internally what methods would be relevant for such a comparison.

Site specific algorithms: There are numerous site (locus) specific algorithms available for analysis of copy number data, but these typically have quite different aims than CARMA and produce a different type of result than the proposed algorithm. A site specific algorithm would generally identify all loci satisfying a particular property, such as increased gene dosage in a single sample or significant increase in copy number across multiple samples. One obvious application of this is to identify candidate drivers of tumor initiation or progression. CARMA, on the other hand, detects the presence of specific motifs throughout the genome or a predefined region in the genome. Our hypothesis was that the latter approach would capture footprints of past and ongoing processes driven by DNA replication and repair errors, and we believe that the results support this hypothesis. Metaphorically speaking, we are thus seeking to identify the smoke rather than the fire itself. As a side note, our approach is both inspired by and conceptually related to the approach used to derive mutational signatures in human cancer (Nik-Zainal et al, Cell 2012; Alexandrov et al., Cell Reports 2013) and later translations

to other molecular levels. Given the above differences in aim, approach and output between CARMA and site specific methods, a direct comparison between CARMA and such methods is difficult. Still, some algorithms are borderline cases and may be considered as hybrids between site specific and non-site specific algorithms. GISTIC is a prominent example of a hybrid method in this respect, as it may report both very narrow regions and broad regions covering a substantial proportion of a chromosome arm. In the revised manuscript we have included a comparison with GISTIC; see below for more details.

Non-site specific algorithms: There are fewer algorithms available for non-site specific analysis of DNA copy number in cancer (several of these are referred to in the paper). One algorithm (Genomic Instability Index; GII) was compared with CARMA in the first version of the manuscript. Inspired by the reviewer's concern, we have identified three additional algorithms (CINdex, CAAI and GISTIC) that we find highly suitable for a comparison with CARMA. These methods capture different types of aberrations and have varying genomic resolutions, thus they span a broad range of alternative approaches. CINdex detects gains and losses at the chromosomal level, CAAI identifies complex rearrangements at chromosome arm level, while GISTIC captures significant copy number hotspots at subchromosomal level.

We have now expanded the method comparison substantially from one method (GII) to four methods (GII, CINdex, CAAI and GISTIC). We have added a new subsection "2.2 Relation to other methods" under Results (page 3-4), a paragraph under Discussion (page 10-11), three new paragraphs in Supplementary Material and Methods (page 7), and a new figure (Figure 2).

In addition, other clinical and pathologically parameters should be included in the analysis, not just survival.

In the four included data sets, the only recorded clinical end point is survival (progression free survival or disease specific survival). On the other hand, we have access to a range of other clinical and pathological parameters: age, T-status, N-status, histological grade, ER status, HER2 status and TP53 mutation status. These are already included in the univariate Cox regression analysis to compare the CARMA derived score CPI and CPI_{weighted} to other variables (Figure 3d). They are also included in the multivariate Cox regression analysis to assess the association between $CPI/CPI_{\text{weighted}}$ and survival when additional factors are adjusted for (Figure 3f).

We have put more emphasis on the latter part of the analysis in the revised manuscript by moving Supplementary Figure S5 to Figure 3f.

Reviewer # 2:

The authors have developed an algorithm named Copy Aberration Regional Mapping Analysis (CARMA) to identify tumour copy number profile based motifs. Then they have proposed to use these motifs to develop a personalized score based on presence/absence of them, which can be used as an independent prognostic predictor for breast cancer. They have compare the CARMA signatures against breast cancer subtypes derived based on PAM50 genes and InClust. Overall, I find the manuscript is well-written and include a good explanation of the method. If CARMA can be used in other cancer types easily and with a good documentation, I think the method will be very useful in many cancer types.

The proposed CARMA method can easily be applied to other cancer types than breast cancer.

This is now explicitly stated in the paper in the Introduction (page 3), and we have also included there a link to a web site with software and detailed documentation explaining how to apply CARMA to new data sets. See also our response to the next question.

The software was not available for testing. It is recommended to have the code and test data set available for testing purposes. So, I couldn't run the programme.

The software is now made available as an R package "carma" for easy installation on any computer from the University of Oslo official web site <http://heim.ifi.uio.no/bioinf/Projects/>. The R package includes detailed documentation, an example data set to make it easy to start using the method, and a suite of functions to summarize the output and to visualize the results both for single samples and for collections of samples. The software is also in the process of being submitted to the repository Comprehensive R Archive Network (CRAN).

The method is mainly devised around calculating six different types of scores (AMP, DEL, etc...). These abbreviations are first given in Figure 1. However, these abbreviations weren't introduced earlier. So, it is difficult to follow the figure and connect the figure with figure references under section 2.1. Please provide the description of these abbreviations before their first appearances.

We have included in the manuscript a description of the abbreviations AMP, DEL, STP, CRV, LOH and ASM in Results under section 2.1 (page 3).

Harrell's C scores are specified to assess the predictive power on survival data. It is recommended to have the confidence interval of these values to have a better understanding of the performance.

We have now included confidence intervals for Harrell's C scores in Results under section 2.4 (page 7).

I couldn't find the reasoning behind the use of arm level scores rather than gene level or chromosome level CARMA scores. Please provide these details for improve the readability.

The CARMA algorithm is not bound to any particular genomic resolution, and the software supports three options for the calculation of scores: chromosome scores, arm scores and bin scores. For bin scores, the genome is subdivided into a number of bins of identical width (the width is specified by the user). Multiple options are provided to handle boundary problems for bin scores (i.e. the fact that an integer number of bins of specified size may not exactly cover a chromosome). In addition, we have included in the software the reporting of scores on the gene level by letting all genes in a given region inherit the region level score. Note, however, that even though the user is free to choose the bin size when that option is selected, the algorithm may fail to capture regional events such as complex rearrangements if the bin size is chosen too small.

We have included a description of the above options in the Discussion (page 10) and we also point out there that the use of arm level scores is just one of multiple options.

Did you only select female samples from the normal samples for the signal normalization? If not, how did you normalize for X chromosome? It is good to have these details in the methods section.

Prior to application of the CARMA algorithm, the raw data (SNP or NGS) have to be preprocessed in order to derive segmented allele-specific copy number estimates. For this step, we have relied on established computational pipelines as described in Supplementary Material and Methods. The selected pipelines are designed to handle samples of mixed genders. The CARMA algorithm itself offers multiple options. The first is to exclude chromosome X from the analysis; the second is to assume that all samples are female, and the third is to provide the gender for each sample. Among the six CARMA indices, only two (AMP and DEL) are sensitive to gender.

A new subsection "3.3 Calculating CARMA scores in sex chromosomes" has been added under Material and Methods to describe how CARMA handles chromosomes X and Y and the available options.

Can CARMA be applied to other cancer types? If yes, please provide some details on this. For example, handling chromosome X and Y scores.

As explained above, CARMA can be applied to any cancer type, and this is now stated in the paper in the Introduction (page 3). Handling of chromosome X and Y scores is also discussed in the revised paper, as explained above.

How CARMA performance comparing against CINdex?

CARMA and CINdex both provide regional scores; however, CARMA provides a much more detailed picture of the regional copy number structure than CINdex, owing to the use of six distinct measures of copy number distortion. For example, in a region with loss of one allele and gain of the other one (i.e. a uniparental disomy), CINdex will not report an alteration while CARMA will report loss of heterozygosity (positive LOH score) and asymmetric allele distribution (positive ASM score). As another example, copy number signatures of complex regional DNA rearrangements such as chromoplexy and chromothripsis will only be reported by CINdex as gain or loss, while CARMA will also report positive steepness (STP) and/or positive curvature (CRV).

A comparison of CARMA and CINdex is now described in Results under section "2.2 Relation to other methods", is further considered in the Discussion (page 10), and is depicted in Figure 2a.

What are the differences between using CARMA to identify CN patterns compared to GISTIC? How CARMA gives better understanding of the CN patterns?

There are several important differences between GISTIC and CARMA (note that by GISTIC we actually refer here and in the paper to the most recent version GISTIC2). GISTIC requires multiple tumor samples (copy number profiles) and seeks to identify genomic regions with significant gain or loss across the samples, while CARMA can be applied to a single tumor sample and detects the presence of six particular copy number motifs (such as a focal complex events) throughout the genome or in predefined regions of the genome. GISTIC generally distinguishes between whole-arm aberrations and focal aberrations, while no such distinction is made in CARMA. GISTIC is based on an estimate of total copy number in each locus, while CARMA is based on allele-specific copy number estimates. One obvious application of GISTIC is to identify regions harboring candidate driver genes for tumor initiation or progression, while CARMA is designed to capture the footprint of past and ongoing copy number alterations caused by e.g. DNA replication and repair errors. In view of these differences, we consider CARMA to be a supplement to rather than a replacement of GISTIC.

The relation between CARMA and GISTIC is described in Results under section "2.2 Relation to other methods" (page 3-4) and is further elaborated in the Discussion (page 10-11). We have also included a new figure (Figure 2a) that directly compares the output from GISTIC and CARMA on two selected samples. We further compared the two methods by showing the relative distributions of the six CARMA scores in each genomic region reported by GISTIC (Figure 2b). This highlights the added value of CARMA which provides a much more detailed picture of the aberration type within each GISTIC region. It also provides a useful additional means for graphically representing the output from CARMA.

Minor comments:

- 1) 'inn' should change to 'in' on page 5 last paragraph.
- 2) What is 't' in page 6? It is not clear in main text.

We have corrected the error in 1) and have clarified 2) in the text (page 8).

Reviewer # 3:

The "DNA copy number motifs are strong and independent predictors of survival in breast cancer" paper hypothesized that short-range copy number features present throughout the genome mirror underlying instability processes. Thus, it uses the copy number features to predict breast cancer. Here, I have the following questions.

First, could author provide a detailed flow chart to describe the CARMA algorithm?

We thank the reviewer for this suggestion.

We have now included a detailed flow chart of the CARMA algorithm in Figure 1. In the revised Figure 1, the overall analysis flow is shown in Figure 1a (as before), the computational flow of the CARMA algorithm is shown in Figure 1b (new), and the detailed computations inside the CARMA algorithm leading to the six regional scores are depicted in Figure 1c (as before). Figure 1d shows three examples (as before).

Second, why the author did not consider the novel single cell sequencing or three-dimensional genome methods to predict breast cancer?

The proposed CARMA algorithm only uses allele-specific copy number data from bulk tumor to detect certain structural features in the tumor DNA. We fully acknowledge that the addition of several layers of information, such as data from single cell sequencing or 3D structural analyses, would allow even more detail to be discerned. For example, with data from single cell sequencing we would be able to study the degree of heterogeneity and subclonality of copy number motifs in a tumor, and with 3D structural information we might be able to explore further causes and consequences of inferred copy number motifs. The reason we chose to consider only allele-specific copy number data from bulk tumor is that such data are (1) already available in many studies; (2) easily and relatively cheaply generated for new samples; and (3) very informative of the motifs that we seek to detect. Having said this, an extension of CARMA to incorporate other data layers would be most interesting.

Third, Authors only use three data sets to demonstrate their exploration and validate the efficiency of the algorithm, which is not enough for a nature affiliated journal.

We agree that a broad validation is valuable and we have identified a fourth breast cancer dataset with copy number data and clinical data available for 553 tumors/patients. This

dataset contains samples collected through the International Cancer Genome Consortium (ICGC) and has previously been published. Both copy number data and clinical data are available from the Supplementary Tables in the original publication (Nik-Zainal, Nature 2016). CARMA analysis of this dataset confirmed the results obtained from the three other cohorts.

For validation, we have included in the paper a fourth breast cancer data set from ICGC. This is described in Results under section "2.4 Predicting survival from regional scores" (page 7), and results from survival analysis is shown in Figure 3e. Detailed information about this dataset has been included in the Supplementary Material and Methods (page 4) and in the section Data Availability (page 14). Supplementary Tables 1 and have been updated to include information about this cohort.

Best wishes,

Ole Christian Lingjærde
Professor, Dept of Informatics, University of Oslo, Norway and Dept of Cancer Genetics, Oslo
University Hospital, Oslo, Norway
Phone: +47 46636659
Email: ole@ifi.uio.no

REVIEWERS' COMMENTS:

Reviewer #1 (Remarks to the Author):

The issues have been addressed in the revision.

Reviewer #2 (Remarks to the Author):

Pladsen et.al. have described a novel algorithm named CARMA in this manuscript. I believe the latest version of the manuscript has addressed the concerns I raised during the last review. The authors have included validation of a novel data set. Also, provided web link to the R package and compared the method against three other methods.

Reviewer #3 (Remarks to the Author):

I am fine with the revision.